# Therapeutic Potential of Thymoquinone in Triple-Negative Breast Cancer Prevention and Progression through the Modulation of the Tumor Microenvironment

**DOI:** 10.3390/nu14010079

**Published:** 2021-12-25

**Authors:** Getinet M. Adinew, Equar Taka, Bereket Mochona, Ramesh B. Badisa, Elizabeth A. Mazzio, Rashid Elhag, Karam F. A. Soliman

**Affiliations:** 1Division of Pharmaceutical Sciences, College of Pharmacy and Pharmaceutical Sciences, Institute of Public Health, Florida A&M University, Tallahassee, FL 32307, USA; getinet1.mequanint@famu.edu (G.M.A.); equar.taka@famu.edu (E.T.); ramesh.badisa@famu.edu (R.B.B.); elizabeth.mazzio@famu.edu (E.A.M.); 2Department of Chemistry, College of Science and Technology, Florida A&M University, Tallahassee, FL 32307, USA; bereket.mochona@famu.edu; 3Department of Biology, College of Science and Technology, Florida A&M University, Tallahassee, FL 32307, USA; rashid.elhag@famu.edu

**Keywords:** breast cancer, tumor-microenvironment, triple-negative breast cancer, thymoquinone

## Abstract

To date, the tumor microenvironment (TME) has gained considerable attention in various areas of cancer research due to its role in driving a loss of immune surveillance and enabling rapid advanced tumor development and progression. The TME plays an integral role in driving advanced aggressive breast cancers, including triple-negative breast cancer (TNBC), a pivotal mediator for tumor cells to communicate with the surrounding cells via lymphatic and circulatory systems. Furthermore, the TME plays a significant role in all steps and stages of carcinogenesis by promoting and stimulating uncontrolled cell proliferation and protecting tumor cells from the immune system. Various cellular components of the TME work together to drive cancer processes, some of which include tumor-associated adipocytes, fibroblasts, macrophages, and neutrophils which sustain perpetual amplification and release of pro-inflammatory molecules such as cytokines. Thymoquinone (TQ), a natural chemical component from black cumin seed, is widely used traditionally and now in clinical trials for the treatment/prevention of multiple types of cancer, showing a potential to mitigate components of TME at various stages by various pathways. In this review, we focus on the role of TME in TNBC cancer progression and the effect of TQ on the TME, emphasizing their anticipated role in the prevention and treatment of TNBC. It was concluded from this review that the multiple components of the TME serve as a critical part of TNBC tumor promotion and stimulation of uncontrolled cell proliferation. Meanwhile, TQ could be a crucial compound in the prevention and progression of TNBC therapy through the modulation of the TME.

## 1. Introduction

Breast cancer (BC) is one of the most life-threatening diseases with the highest number of human cancer subtypes and is the second leading cause of death in women [1]. In 2020, there were approximately 2.3 million new breast cancer cases and 685,000 breast cancer deaths globally [2], with 1,898,160 new cases and 608,570 deaths projected in the United States by the end of 2021 [3]. Triple-negative breast cancer (TNBC) is characterized by a lack of expressed progesterone receptors, estrogen receptors, and human epidermal growth factor 2 (HER2) [4]. It is one of the major metastatic, drug-resistant, and aggressive sub-classes of BC [5], which comprises around 20% of total BC globally [6], and 83% of deaths compared with other BC subtypes [7]. There are various factors involved in the initiation and progression of TNBC. The most indispensable is the TME, directly linked with angiogenesis, rapid cell proliferation, immune system suppression, apoptosis inhibition, and eventual evasion of immune surveillance in the tumor area [8]. Because of these activities, TME can be noted as a critical hallmark in cancer pathogenicity and a potential target for TNBC [9]. 

Even though combination chemotherapy with taxanes and anthracyclines is the main treatment of TNBC [10], mortality rates are associated with a heterogeneity of oncogenic drivers, chemoresistance, systemic toxicity, and poor selectivity, establishing challenging limiting factors [11]. Side effects of conventional therapy for BC patients involve impaired fertility, osteoporosis, premature menopause, congestive heart failure, cardiomyopathy, arthralgia, and myalgia [12]. Given that TNBC is one of the most common life-threatening and complex kinds of BC, substantial research has been devoted to discovering novel biomarkers and biologically targeted treatment sites to enhance patient prognosis and clinical outcomes compared to other subtypes [13]. 

## 2. Tumor Microenvironment (TME) and TNBC

The tumor microenvironment is crucial for the induction and development of TNBC. The initiation of angiogenesis, proliferation, apoptosis inhibition, immune system suppression, and evasion of immune surveillance is inherently linked to TME [8]. Naturally, TME is heterogeneous and highly complex [14], enabling cancer cells’ rapid proliferation, subsequent development of hypoxia, and concomitant reprogramming of cancer cells in the TME to acclimatize to changes within TME. The TME generates a dwelling place for interacting cancer cells with their neighboring immune and endothelial cells and fibroblasts and can provide potential targets for TNBC [9]. 

The reciprocal communication between stromal cells and cancer cells produces changes in the cellular elements of TME, which predisposes tumor cells to metastasis [15]. Cancer cells generally gain a supportive microenvironment by activating the wound healing response of the host [16]. Morphologically, the TME resembles a wound healing site [17], composed of different histological changes, including inflammatory cell infiltration, extravascular clotting, accumulation of activated fibroblast, angiogenesis, and synthesis of extracellular matrix [18]. Likewise, the stromal cells, such as tumor-associated macrophages or cancer-associated fibroblasts, promote tumor progression by generating growth factors, chemokines, and promigratory extracellular matrix components [19]. Additionally, TME stimulates the transition of epithelial cells to TNBC stem cells [20]. A mounting number of studies have focused on the TME to discover the complex mechanisms underlying tumorigenesis, elucidate new biomarkers and drug targets that may predict clinical outcomes, and guide therapy in TNBCs [21]. The various components in the microenvironment promote tumorigenesis, protect tumor cells from host immunity, and reduce chemotherapeutic response effectiveness [22] (Figure 1). 

## 3. Natural Products Targeting TME

Natural products have been utilized for centuries in traditional medicines, ubiquitously distributed throughout plants and marine organisms [23]. The majority of bioactive natural products are secondary metabolites that have been isolated and experimentally validated for their anti-bacterial, anti-viral, anti-fungal, anti-inflammatory, and anti-cancer properties [24,25]. For many decades there has been growing interest in natural compounds with anti-cancer activities since they are less/nontoxic, inexpensive, and easily accessible by most people, which is preferable for the prevention and treatment of various ailments [26]. Around 45% of all anti-cancer drugs in clinical use originate from natural compound secondary metabolites [27,28]. Natural compounds exhibit a significant effect on the tumor cells via the modulation of TME [29] and research directions have now shifted toward natural compounds and specific TME-modulating properties [30,31,32]. Considering the data emerging from these efforts, it has become advisable that we revisit TQ to understand its broader significance in the modulation of TME and associated mechanisms. The expectation is that the current review would encourage future translational advancements of this compound in clinical use to prevent and treat TNBC. 

## 4. Thymoquinone (TQ)

The Nigella sativa seed, also known as black seed, is commonly used in different areas globally to prevent and treat diseases [33]. It contains various bioactive compounds, such as thymoquinone, p-cymene, α-piene, thymo-hydroquinone, dithymo-quinone, and t-anethol. These compounds possess several pharmacological properties, including anti-neoplastic, anti-inflammatory, antioxidant, anti-asthmatic, analgesic, antipyretic, antihypertensive, and antimicrobial [34,35]. Among these constituents, TQ has shown a promising role against various diseases, including cancer, such as breast cancer, prostate cancer, bone cancer, gastric cancer, bladder cancer, colon cancer, and lung cancer [26,36]. 

TQ can protect the healthy cells from oxidative damage and provide long recovery for cells by preventing toxic side effects [37]. TQ exhibits antiproliferative effects on different cancer cell lines of the breast, ovary, larynx, colon, myeloblastic leukemia, osteosarcoma, and lung [38,39]. In BC models, TQ can lower VEGF, enhance serum INF-γ levels, suppress angiogenesis, and shift the immune response toward T helper1 [40]. It has also been reported that by targeting cyclin E, cyclin D1, and p27 proteins, TQ caused cell cycle arrest, suppressing progression from G1 to S phase [41]. TQ can also induce apoptosis by activating caspase-8, cleaving poly (ADP-ribose) polymerase, and lowering AKT phosphorylation [42,43]. Moreover, TQ is effective in xenograft tumor models through multiple mechanisms, including inducing p38 phosphorylation and inhibiting protein expression of anti-apoptotic genes, i.e., Bcl-2, Bcl-Xl XIAP, and survivin. These findings are fairly consistent throughout the literature, where TQ enhances TUNEL, lowers Ki67 [44,45], activates caspase-9, induces apoptotic cell death, and interferes with the survival of cells involving the PI3-K/Akt pathway [46,47]. Figure 2 describes the possible anti-cancer mechanisms of TQ. 

## 5. Thymoquinone (TQ) Targeting the Cellular Components of TME

The cellular components and secretary molecules of TME targeted by TQ as discussed here are tumor-associated macrophage(TAM), cancer-associated fibroblast(CAF), cancer-associated adipocytes(CAA), cholesterol synthesis, reactive oxygen species (ROS), eukaryotic elongation factor-2 kinase (eEF-2K), tumor-infiltrating lymphocytes (TILs), indoleamine 2,3-dioxygenase (IDO), vascular endothelial growth factor (VEGF), transforming growth factor-β (TGF-β), insulin-like growth factors (IGF-I), endoglin, IL-6, and JAK/STAT signaling. 

### 5.1. Effect of TQ on Cholesterol Synthesis and Its Metabolites

Cholesterol is a crucial lipid molecule in normal physiological processes, being critical for intracellular signal transduction, an essential component of the plasma membrane, and a precursor for steroid hormone [48]. However, the accumulation of cholesterol is associated with increased cancer cell proliferation in different types of cancer, including TNBC [49]. Because cancer cells proliferate rapidly, they rely heavily on cholesterol to meet their growing demand for substrates for membrane production [50]. As a result, cholesterol is helpful for cancer growth and development because it increases oncogenic signaling, apoptosis evasion, cell motility, and invasion [51,52]. Cancer cells, for example, increase their cholesterol requirement by boosting de novo production or absorption, modifying cholesterol outflow or increasing cholesterol storage [53]. Cholesterol and its metabolites produced by tumor cells have influenced the phenotype and function of cells forming the TME. Increased glucose uptake and de novo steroidogenesis principally satisfy the high nutrient demand of tumor cells. However, increased cholesterol metabolism and de novo lipogenesis are also detrimental for the cells forming the TME, such as immune cells [54].

Several oncogenic signals include steroidogenic acute regulatory protein (STAR), STAR-related lipid transfer domain containing 3 (STARD_3_), RTK/RAS, PI_3_K/AKT/mTOR, and TP_53,_ have been shown to modulate cholesterol synthesis in cancer cells. Overexpression of STAR and STARD_3_ increases cholesterol biosynthesis by increasing the expression of cholesterol synthesis enzymes [55]. These proteins are elevated in cancer cells and resistant to apoptotic signals [56]. While STAR and STARD3 are upregulated in BC cells, including TNBC, genetic knockdown of these genes increases sensitivity to chemotherapeutic agents, reduced cell proliferation, and increases apoptosis [55,56,57]. Constitutive upregulation of PI_3_K/AKT signaling increases intracellular cholesterol levels by activating sterol responsive element binding protein (SREBP), which enables induction of the LDL (low-density lipoprotein) receptor that mediates cholesterol import. It decreases ABCA1, which mediates cholesterol export in a mTORC_1_-dependent manner [58]. Evidence indicates that the PI_3_K/AKT/mTOR pathway contributes to cancer cell growth and progression [59]. Targeting these genes either by inducing p53 or inhibited STAR and STARD_3_, PI_3_K/AKT/mTOR, RTK/RAS or targeting the mevalonate–cholesterol biosynthesis pathway to regulate cholesterol synthesis and its metabolites using TQ will be an alternative option in the prevention of TNBC, either alone or in combination with other chemotherapeutic agents.

Moreover, TP_53_, the most frequently mutated gene in TNBC cells, is another gene deregulating the cholesterol synthesis pathway. Through upregulation of the mevalonate pathway, an essential pathway in the synthesis of cholesterol in cancer cells, tumor cells continuously increase activation of cholesterol synthesis. Any agents that inhibited the mevalonate pathway or downregulated mutant P_53_ are a potential target for preventing TNBC via modulation of cholesterol synthesis [60]. 

TNBC cells display distinct adipogenic characteristics in the mevalonate–cholesterol biosynthesis pathway and metabolic drug responses. In a mouse MDAMB-468 xenograft tumor model, co-administration of 2-DG plus gefitinib significantly reduced tumor size. G28, a fatty acid synthase (FASN) inhibitor, had a substantial antiproliferative effect on the TNBC cell line MDA-MB-231) and its doxorubicin (231DXR) and paclitaxel-resistant variants (231PTR) [61]. EGCG, an anti-FASN compound, in combination with cetuximab, showed potent anticancer efficacy in doxorubicin-resistant TNBC cell lines (231DXR and HCCDXR) [62]. 

The cholesteryl ester-producing enzyme acetyltransferase (ACAT-1) is overexpressed in two TNBC cell lines of human breast cancer, MDA-MB-231, and MDA-MB-436. A faster cell growth rate could be connected to higher enzyme levels. This study was confirmed when TNBC cells were treated with CP-113818, an ACAT-1 inhibitor, which inhibited cell proliferation and migration by modulating cholesterol metabolism [63].

According to in vivo and in vitro studies, phosphatase and tensin homolog (PTEN) loss promotes tumorigenesis and metastatic potential in pancreatic cancer by activating the downstream PI3K/Akt/mTOR/SREBP signaling pathway; higher levels of esterified cholesterol boost carcinogenesis and metastatic potential [64]. Several studies show that TQ induces cell death by inhibiting NF-kβ/Akt activation via upregulation of PTEN. This effect leads to phosphorylation of Akt and induction of p53 protein and its transcriptional target p21, thus evoking G2/M phase arrest and apoptosis in drug-resistant hormone-positive BC cells [46]. Another study looked at the role of TQ in different cells, finding that it regulated several target proteins involved in the PI3K/AKT signaling network. TQ lowered PTEN phosphorylation at Ser380 (PTEN inactivation) in the MDA-MB-468 TNBC cell line, resulting in PTEN activation [65]. In the MCF-7 breast cancer cell line, TQ promotes apoptosis by upregulating P53 expression in a time-dependent manner [66]. 

TQ will be an alternative option in the prevention of TNBC, either alone or in combination with other chemotherapeutic agents, by inducing p53 or upregulation of PTEN or inhibiting STAR and STARD3, PI3K/AKT/mTOR, RTK/RAS, or targeting the mevalonate–cholesterol biosynthesis pathway to regulate cholesterol synthesis and its metabolites.

### 5.2. Effect of TQ on Reactive Oxygen Species (ROS)

Reactive oxygen species (ROS) are vigorously reactive oxygen molecules. They are produced in cells during normal mitochondrial oxidative metabolism and cellular respiration in response to xenobiotics. ROS can oxidatively damage the cellular components such as carbohydrates, lipids, proteins, and nucleic acids and change their function [67]. They are considered a key factor for many pathological conditions, including cancer. ROS contribute to several alterations of physiological function, structure, and mutagenesis in DNA, thus producing cancer [68]. ROS within a cell plays an essential role in controlling cell proliferation, apoptosis, and metabolic pathways [69]. ROS are produced by cancer cells and cellular components in the microenvironment via disturbed metabolism of the cancer cells [70]. They directly change the TME by activating cancer cells and inflammatory cells, which stimulates various inflammatory molecules to promote carcinogenesis [71]. Therefore, they are recognized as the hallmarks of cancer secreted in the tumor-promoting microenvironment [8]. 

ROS are master modulator of tumorigenesis [72]. Many cancer processes are redox-sensitive. These include proliferation, angiogenesis, cell motility, cell cycle progression, cell survival and apoptosis, cell morphology, energy metabolism, cell-to-cell adhesion, and tumor stemness [73]. Furthermore, it has been broadly described that oncogene activation promotes ROS accumulation in tumor cells [74]. It has been reported that ROS has dual actions, both tumor-suppressive or oncogenic functions, as various studies have shown contradictory results [75]. The functions are dependent on cancer stages, levels of ROS, and other trigger factors in cancer cells [76]. When ROS are high, they increase apoptosis, senescence, cellular damage, trigger microsatellite instability via DNA damage induction [77,78,79,80,81], and suppress tumorigenesis through the p38 pathway. Conversely, low to moderate levels of ROS promote proliferation via activation of various kinase cascades such as mitogen-activated protein kinase (MAPK)/extracellular signal-regulated kinase (ERK)-Ras-Raf-MEK-ERK pathway phosphoinositide-3-kinase/protein kinase B (PI3K/Akt), cyclin D1 expression, and c-Jun N-terminal kinase (JNK) [82]. Moreover, ROS promotes the reversible deactivation of PTEN [83].

Several studies have reported that natural compounds such as catechins exert an antioxidant effect by directly scavenging free radicals through donating one electron of their phenolic group, and via chelating metal ions involved in radical production. Indirectly, catechins enhance the activity of antioxidant enzymes such as catalase, superoxide dismutase and glutathione peroxidase, and can inhibit pro-oxidant enzymes [84]. On the contrary, authors looking at Syringin induce oxidative stress to suppress the growth and proliferation of TNBC cells; however, the antioxidant N-acetylcysteine reversed the ROS levels and prevented apoptosis [85]. Similar studies have reported that ROS activates the antitumor activities of doxorubicin, resulting in the efficient inhibition of distant metastasis and tumor growth in TNBC cells [86]. 

Meanwhile, TQ has dual functions towards the regulation of ROS. Evidence shows that TQ inhibits proliferation and promotes apoptotic effects through upregulation of ROS in BC cells both in vivo and in vitro models [44,87]. The authors divulged TQ’s positive impact on ROS production and tumor suppression. Downregulation of XIAP(X-linked inhibitor of apoptosis protein), survivin, Bcl-xL, and Bcl-2 protein, as well as accelerating oxidative stress, can cause oxidative damage in mitochondria, triggering apoptosis [88]. 

On the other hand, TQ acts as antioxidant activity by inducing cytoprotective enzymes resulting in cells’ protection against ROS-induced cellular damage. Studies have reported that TQ upregulates cytoprotective enzymes, including superoxide dismutase, catalase, glutathione-S-transferase, glutathione reductase, and glutathione peroxidase, and functions by scavenging superoxide radicals and hydrogen peroxide and inhibiting lipid peroxidation [89,90,91]. TQ can also relay its antioxidant activity through increased activation of Nrf2 (nuclear factor-erythroid 2 related factor 2) in the nucleus, thus upregulating ARE gene activity [92]. 

These data suggest that TQ has anti-cancer activities via modulation of TME through suppression of ROS in TNBC, as shown in Figure 3.

### 5.3. Effect of TQ on Eukaryotic Elongation Factor-2 Kinase (eEF-2K) 

Eukaryotic elongation factor-2(eEF-2K) is the only known substrate for eEF-2kinase that promotes ribosomal translocation from A-site to P-site [94]. eEF-2k is a Ca^2+^/CaM-dependent member of the kinase family function, by phosphorylating/inactivating eEF-2K at threonine 56, leading to reduced peptide chain elongating [95], then increasing eEF-2K function, promoting tumor cell survival, growth, and metastasis [96]. Activation of eEF2K by oxidized low-density lipoprotein inhibits apoptosis in macrophages and promotes survival of cancer cells [97]. eEF-2K increases tumor necrosis factor-alpha (TNF-α), promotes TAM survival and function, induces nitric oxide (NO) in activated macrophages, and contributes to aggressive tumor behaviors [98]. eEF2K expression has been shown to protect cancer cells from hypoxia [99] by producing the expression of specific proteins, such as microtubule-associated proteins, resulting in the expression of tumor-promoting proteins and growth factors [100]. 

The TME is highly acidic and more hypoxic than normal tissues, leading to tumor aggressiveness [101]. eEF2K was proposed to be responsible for cell survival under acidic [102] and hypoxic conditions [99] in cancer cells and fibroblasts. Taken together, studies highlight the importance of eEF2K in both cancer cells and microenvironmental cells. Natural compounds have been used to target eEF2K via different mechanisms, such as interfering with the eEF2-binding site, CaM-binding site, or the ATP-binding site. For instance, rottlerin, a natural compound, acts as a non-specific inhibitor of eEF2K [103]. Studies showed that rottlerin suppressed the eEF2K gene and protein expression in cancer cells [104]. 

Studies have shown that eEF-2K significantly increased in TNBC cells associated with poor patient survival and prognosis, promoting cell proliferation, invasion, migration, chemotherapy resistance, and angiogenesis. In vivo studies of a highly aggressive TNBC MDA-MB-231 tumor in an orthotopic xenograft mice model, eEF-2K, were significantly elevated. This was confirmed by liposomal eEF-2K siRNA delivery, which resulted in down-regulation [105]. Similar studies also revealed that the loss/reduced expression of tumor suppressor gene-miR-603- leads to overexpression of eEF2K, and at the same time, when miR-360 is upregulated, results in downregulation of eEF2k by directly targeting the 3-UTR, then inhibits tumor cell migration, growth, and invasion in TNBC [106]. Overexpression of miR-603, which suppresses eEF-2k and inhibits cell migration, proliferation, and invasion in TQ-treated TNBC in mice models and the MDA-MB-231 cell line, has been found to reduce cell migration, proliferation, and invasion by targeting the miR-603/eEF-2k axis [107]. These findings show that utilizing TQ to prevent and treat TNBC that targets eEF2k could be a viable approach (Figure 4).

### 5.4. Effect of TQ on Inflammatory and Immune Cells 

Inflammatory/immune cells in the TME play a role in cancer cell survival, development, proliferation, and differentiation by boosting growth and survival factors and encouraging angiogenesis, inducing mutation by damaging DNA and interfering with treatment [108]. In the TME, the prominent inflammatory and immune cells are tumor-associated macrophage (TAM), cancer-associated adipocytes (CAAs), cancer-associated fibroblasts (CAFs), tumor-infiltrating lymphocytes (TILs), tumor-associated neutrophils (TANs), myeloid-derived suppressor cells (MDSCs), T cells, B cells, natural killer (NK) cells, and dendritic cells (DCs). The cytokines and other factors and signaling molecules produced by these cells include interleukin-1 beta (IL-1β), interleukin-6 (IL-6), interleukin-10 (IL-10), interleukin-17 (IL-17), tumor necrosis factor-α (TNF-α), vascular endothelial growth factor (VEGF), transforming growth factor-β (TGF-β), and matrix metalloproteinases (MMPs). This review particularly focuses on the roles of TAMs, CAAs, CAFs, TILs, Il6, and TGF-β in the TME

#### 5.4.1. Effect of TQ on Tumor-Associated Macrophage (TAM)

Macrophages are phagocytes that serve as a first-line defense against a disease that insults tissue. They are recruited in large numbers to tumors, resulting in immunosuppression. TAMs are derived from circulating monocytic precursors and are directed into tumors by chemokines. When TAMs are appropriately activated, they can kill tumor cells, but persistent expression causes tumorigenesis [109]. TAMs are related to the high expression of cytokines and chemokines that promote tumor growth progression, tumor cell proliferation, angiogenesis, favoring metastasis and invasion, and suppressing antitumor immunity [110].

TAMs are the most abundant immune cells and account for 30–50% of the total inflammatory cells in the TME. TAMs alter the TME by encouraging the proliferation of tumor cells via enrichment of the cytokines and growth factors and thus likely promote, rather than inhibit, cancer development and progression in TNBC [111]. In tumor cells, high TAM numbers are strongly associated with increased vascular density [112], inflammatory cytokines, and growth factors that help angiogenesis by upregulating endothelial cell survival, activation, and proliferation, and are also an important source of VEGF-A both in mice and human tumors [113]. VEGF-A derived from TAM also promotes vascular permeability, facilitating cancer cell intravasation and metastasis [114]. Evidence shows that VEGF-A deficiency in TAMs limits their ability to restore angiogenesis and relapse transplanted tumors after chemotherapy [115].

TAMs favor cancer cell tumor development, invasion, and metastasis into distant sites. TAMs promoted MDA-MB-231 TNBC cell proliferation through activating phosphoinositide 3-kinase (PI3K)-Akt signaling, as well as apoptosis inhibition through increased Bcl-2 and decreased Bax expression, according to in vitro studies. The effect was validated by suppressing PI3K-Akt signaling by adenoviral siRNA Akt1 transfection, which decreased PI3K-Akt signaling [116]. Furthermore, TAM recruited to the lung, which promotes tumor cell seeding and growth by uniquely expressing FMS-like tyrosine kinase 1 (Flt1, also known as VEGFR1), labels a subset of macrophages in human breast cancers that are significantly enriched in metastatic sites, according to studies using the murine TNBC cell line E0771 [117].

TAMs stimulate tumor angiogenesis, matrix remodeling, tumor cell migration and invasion, and immunosuppression in TNBC by secreting inhibitory cytokines, reducing the effector activities of Tumor-Infiltrating Lymphocytes (TILs), and increasing regulatory T cell proliferation, inflammatory molecules, and M2 markers [118]. Studies elucidated the mutual connection between TAMs and TNBC cells. In comparison to macrophages co-cultured with HR+ BC T47D cells, macrophages co-cultured with TNBC MDA-MB-231 cells had a different morphology, more aggressive behavior, higher levels of macrophage mannose receptor, and other M2 markers, and pro-inflammatory mediators such CCL2 [119].

Thus, TAMs and their downstream molecules may be attractive target sites for novel anti-cancer therapies. The best target would be to identify common proteins expressed or overexpressed only by TAM, and not by resident macrophages of distant, healthy cells, which are essential to face pathogens and could take part in anti-cancer actions. Considering this, chemokines and chemokine receptors, and pro-angiogenic factors are the potential target site for TNBC treatment. For instance, in BC, malignant cells recruit macrophages via the chemokine CCL5, and treatment with its receptor antagonist leads to a decreased number of infiltrating macrophages associated with a significantly reduced tumor size [120]. TAM pro-angiogenic functions have also been considered as another target site for anti-cancer development. Antiangiogenic factors have been demonstrated to inhibit macrophage recruitment and reduce tumor size [121]. Furthermore, in an orthotopic breast tumor model, anti-VEGF-A neutralizing antibody therapy reduced the creation of new blood and lymphatic vessels and the incidence of lymphatic and pulmonary metastasis [122]. VEGF promotes macrophage recruitment into tumors, and studies demonstrated that specific inhibition of VEGFR2 decreases tumor macrophage infiltration significantly [123].

TQ has been shown to have considerable anti-inflammatory effects on macrophages after oral administration by suppressing inducible nitric oxide synthase enzymes in response to inflammatory stimuli in mice [124]. According to research, TQ lowered tumor NF-β activity, M2 macrophages, and soluble VEGF levels, reducing proliferation and enhancing apoptosis in the ID8-NGL mouse model of ovarian cancer [125]. TQ’s anti-angiogenic activities were tested in a zebrafish angiogenesis model in vivo. The anti-angiogenic action of TQ was examined in zebrafish embryos by testing its inhibitory influence on blood vessel formation and identifying its molecular target. TQ decreased the formation of intersegmental vessels in zebrafish embryos in a dose-dependent manner, according to the findings. TQ was also discovered to suppress VEGF-A mRNA expression [126]. TQ has also been demonstrated to reduce cell proliferation, migration, invasion, and metastasis in TNBC by downregulating TAM-related chemokine receptor expression [127]. According to the findings, TQ may have an anti-cancer effect by modifying proteins that are either overexpressed or under-expressed by TAMs, indicating that it could be used to prevent and treat TNBC. The overall impact of TAM on TNBC carcinogenesis and the role of TQ is shown in Figure 5. 

#### 5.4.2. Effect of TQ on Cancer-Associated Adipocytes (CAAs)

Adipose tissues are the basic supportive site of the energy storage depot. They have an endocrine function producing hormones, growth factors, cytokines, and other molecules named adipokines that regulate energy and metabolic homeostasis via various signaling processes [128,129] stimulate survival and growth of MDA-MB-231 TNBC cell [130]. One of the significant and primary cellular components of TME in BC is the adipose tissue that surrounds it and is involved in tumor initiation, growth, and metastasis via the promotion of angiogenesis [131]. It has been reported that tumor growth was increased when tumor cells were implanted in the brown or white tissue compared with the subcutaneous space of BC mice [132]. Consistent with these findings, the peritumoral adipose tissue is significantly vascularized and macrophage-rich. It produces higher proteases levels than normal adipose tissue [132,133]. 

Researchers have been examining how adipocytes interact with TNBC cells, since they are one of the most prominent resident cells in TNBC’s TME [134]. The study used TNBC model 4T1 cells from mouse mammary cancer and adipocytes. Fatty acid-binding protein 4, a hallmark exclusively expressed at high levels in mature adipocytes, was significantly downregulated, while cytokine CCL2, a potent chemokine for monocyte recruitment, was significantly upregulated in mature adipocytes cultured in a transwell system with or without 4T1 murine TNBC cells [135]. These findings suggest that TAAs have a role in TNBC progression and worsen the immunosuppressive TME by secreting an excessive amount of CCL2. 

CAAs stimulates the secretion of various inflammatory factors such as chemokine (C-C motif) ligand 2(CCL2), CCL5, IL-1B, IL-6, TNF-α, VEGF, and leptin that contribute to promoting the metastasis and invasion of BC [136]. It has been reported that when TNBC cells (MDA-MB-231) were co-cultured with human adipocytes, studies showed that they boosted motility and invasiveness. The addition of glucose, which is utilized to stimulate the function of human adipocytes, furthered this effect [137]. It has been reported that adipokines produced by adipocytes, which secrete ILs, TNF-α, and hepatic growth factor, support the growth of BC by transferring fatty acids to enhance mitochondrial oxidation, extracellular matrix remodeling, and endotrophin production from collagen, supporting the growth of BC by transferring fatty acids to enhance mitochondrial oxidation, extracellular matrix remodeling, and endotrophin production from collagen [138]. Furthermore, adipocytes express programmed death-ligand 1 (PD-L1), which inhibits CD8+ T cell antitumor function; this was validated by an adipogenesis inhibitor, which reduces PD-L1 expression, enhancing the immune system’s ability to fight cancer [139]. 

TNBC treatment may target inflammatory factors, pro-angiogenic proteins, transporters, chemokines, cytokines, interleukins, and other downstream signaling molecules and factors released by CAAs. As a result, inhibiting these molecules offers significant promise for modifying the immunosuppressive TME and allowing for greater immunotherapy in the treatment of TNBC. TQ has been shown to have anti-inflammatory and immunomodulatory properties in both cellular and humoral immunity [140], as well as antiangiogenic effects [126].

According to a comprehensive review of studies, TQ has also been proven to protect against inflammatory factors by lowering pro-inflammatory cytokines such as IL-1, IL-6, and TNF-α, which are also produced by CAAs [141]. Because thymoquinone reduces the synthesis of these damaging pro-inflammatory cytokines, it also reduces the creation of free radicals in the TME. Furthermore, in vivo animal investigations demonstrated that TQ reduced adipocyte hypertrophy-induced inflammation [142], implying that TQ could be an alternative option to prevent TNBC associated with cancer-induced adipocytes in the microenvironment. 

#### 5.4.3. Effect of TQ on Cancer-Associated Fibroblasts (CAFs)

Fibroblasts are typically spindle-shaped cells with a flat oval nucleus located in tissues that synthesize collagen and other extracellular matrix proteins, providing a scaffold for cells, and playing significant roles in determining cell function and phenotype [143]. Fibroblasts provide oncogenic signals to malignant cells and to the transformed epithelia in a paracrine fashion. Carcinoma-associated fibroblast is an activated fibroblast, also known as myofibroblast, found in association with cancer cells which might facilitate angiogenesis and cancer progression [144]. CAFs in cancer cells promotes angiogenesis, decrease cancer cell dormancy and accelerate tumor growth in mice [145]. They are a significant source of tumor VEGFA [146] and support tumor angiogenesis in a VEGFA independent manner [147]. CAF-derived PDGFC (platelet-derived growth factor C) maintains angiogenesis by further secreting CAFs to stimulate pro-angiogenic growth factors, such as FGF2 (fibroblast growth factor) and osteopontin [148]. They indirectly modulate vascularization and blood flow in tumors, secrete enzymes such as lysyl oxidases and hydroxylases to catalyze the crosslinking of collagen to elastin and extracellular matrix molecules [149], and potentiate tumor angiogenesis by attracting vascular endothelial cells and recruiting monocytes from the bone marrow [150].

An in vitro investigation used patients’ specimens with primary invasive TNBC and BT-549, MDA-MB-231, and MDA-MB-468 cell lines cocultured with CAFs. According to their findings, CAFs enhance TNBC cell migration, invasion, and proliferation and participate in the EMT process via the Wnt/β-catenin pathway [151]. Other studies found that CAFs with high expression of leucine-rich repeat-containing protein 15 (LRRC15) increased cell migration and invasion by influencing the Wnt/β-catenin signaling pathway using MDA-MB-231 and MDA-MB-468 TNBC cell lines [152].

An in vivo animal model study reported that increased collagen expression is strongly associated with TNBC, and transforming growth factor-β (TGF-β) ligands produced by cancer cells and/or CAFs promote the accumulation of fibrotic desmoplastic tissues and the rate of cancer progression [153]. TGF- plays a critical role in TNBC epirubicin resistance by modulating stemness, EMT, and apoptosis in epirubicin-treated MDA-MB-231 TNBC cell lines, according to a study [154]. This was further supported by studies conducted both in vitro and in vivo that showed that TNBC tumors with enhanced collagen expression were reversed/prevented by pirfenidone, an antifibrotic agent that antagonizes TGF-β in patient-derived xenograft and 4T1 TNBC homograft tumors models [155]. TQ inhibited platelet-derived growth factor-BB (PDGF-BB), which resulted in lower proliferation and migration of vascular smooth muscle cells, according to a study conducted in vitro and in vivo [156]. TQ concentration-dependently inhibited several growth factors such as EGF and VEGF, the main CAF sources, in breast cancer cell lines in Balb/C mice [157]. In vitro studies on MDA-MB-231 cancer stem cells, TQ inhibited Wnt3a and PI3K and blunted the stimulatory effects of VEGF, EGF, and FGF. The study was further confirmed by demonstrating a lack of cellular response to pro-angiogenesis factors [158]. These data suggest that targeting CAFs are being considered to control TNBC using TQ.

#### 5.4.4. Effect of TQ on Tumor-Infiltrating Lymphocytes (TILs)

TILs consist of T cells and are a significant player in breast TME [159], in which TNBC shows a higher rate of TIL infiltration within the TME [160]. Consequently, higher TIL tumors are more likely to be associated with estrogen negative [161]. Studies confirmed that TILs are most frequently found in highly proliferative tumors such as TNBC than in hormone-positive BC [162]. Clinical trials have demonstrated a strong association between TIL and TNBC. The excellent overall survival and disease-free survival rate were better for early-stage TNBC patients having higher expression of TILs after adjuvant therapy than low levels. This suggested that TILs improved outcomes, and TILs at diagnosis likely indicate an ongoing antitumor immune response [163]. In human and mouse datasets, activation of the Ras/MAPK pathway is significantly linked with a low level of TIL [164].

There is no information on how TQ affects TIL in TNBC. However, several investigations have shown that TQ improves the immune system by improving the immunological response to malignancies. T lymphocyte cell depletion was detected in rats after exposure to gamma radiation, which was recovered by the injection of TQ in mice by raising the percentage of CD4+ and CD8+ cells through modifying Bcl-2, Bax, PD-1, and caspase-3 expression [165]. When compared to the placebo or baseline group, N. Sativa oil alleviated autoimmune disease by boosting the expression of CD4+ cells in a clinical trial. TQ may be engaged in anti-inflammatory effects in TNBC patients by increasing the activity of FoxP3+ Treg cells [166]. TQ dramatically reduced diabetes complications in Streptozotocin-induced gestational diabetic mice by boosting T-cell proliferation and restoring thymus homing CD4+, CD8+ and circulating T-cells [167]. Foxp3+, CD3+, CD4+ and CD8+ T lymphocytes are the broadly studied major components of TILs [168]. Several pre-clinical and clinical trial studies supported that these high levels of TILs present at diagnosis were strongly associated with reduced distant recurrence rates and favorably altered the immune microenvironment [162,163,169]. This finding backs up TQ’s immunomodulatory properties in TNBC patients, suggesting that TQ will be important in increasing TILs through stimulating their core components such as Foxp3+, CD3+, CD4+, and CD8+.

#### 5.4.5. Effect of TQ on Interleukin-6 (IL-6)

Inflammatory TME is regulated by several cytokines [170], including interleukin (IL)-1 and IL-6, which promote cancer cell proliferation and invasion, increasing intracellular signaling by NF-kB, and cytokine receptor activation accelerates tumor progression [171]. IL-6 is produced by several typical cell types, including monocytes, fibroblasts, and lymphocytes, and an autocrine production molecule that occurs in immune cells acts in a paracrine condition to promote neovascularization and inflammation-dependent carcinogenesis [172]. IL-6 participates in several biological activities, including immune regulation, hematopoiesis, and tumorigenesis [173]. An upregulated level of IL-6 in the TME is strongly associated with cancer cell proliferation, epithelial to mesenchymal transition, chemoresistance, invasion, metastasis survival of tumor cells, and angiogenesis through fueling STAT3 MAPK and Akt signaling [174]. Overexpression of IL-6 and its receptors in TME might be due to the increased cell proliferation by increasing Bcl-2 expression, which leads to changes in the proliferation/apoptosis balance towards neoplastic cell proliferation [175]. 

IL-6 is critical for the growth of TNBC, anchorage-independent colony formation, and resistance to apoptosis which contributes to poor prognosis in TNBC patients [176]. IL-6 is secreted by TNBC that increases expression of CCL5 in lymphatic endothelial cells by activating the IL-6 receptor and subsequently STAT3, which increases transcription of the CCL5 gene [177]. IL-6 is expressed in approximately 50% of BC, and its serum level was higher in TNBC than in hormone-positive BC patients [178]. To initiate signal transduction, IL-6 with IL-6 receptor recruits glycoprotein 130 kDa (GP130), a transmembrane protein, also called CD130, ubiquitously expressed in various tumor cells. This is a common receptor for well-known cytokines such as IL-6, 11, 27, LIF, CNTF, OSM, CT-1, and CLC) [179] forming a hexameric IL-6/IL-6R/GP130 complex, which further activates the downstream effector cascade, including phosphorylation of STAT3 [174]. These effects promote tumor survival, proliferation, differentiation, angiogenesis, metastasis, immunosuppression, apoptosis, and drug resistance and advance many human cancers [180]. 

Studies conducted using four different TNBC cell lines (SUM159, MDA-MB-231, MDA-MB-468, MDA-MB-436) have shown that inhibition of IL-6 expression by shRNA in TNBC leads to reduced cell survival and suppression of colony formation both in vivo and in vitro [176]. Similar studies reported that inhibitor of IL-6 receptor signaling strongly inhibits tumor growth and metastasis in TNBC cells (MDA-MB-231, SUM149, and SUM159 cells) [181]. TNBC cells mainly express the transmembrane GP130/IL-6 receptor, and developed specific molecule inhibitors targeting GP130 might be an alternative strategy for effective treatment or prevention of TNBC either in combination with chemotherapy or as a single agent; there are previous studies that have confirmed this scenario [182,183,184]. It has been reported that TQ downregulated IL6 induced phosphorylation of STAT3 and Akt in a concentration and time-dependent manner [185]; hence IL-6 activated Akt has been shown to play a major role in the mechanism of action of IL-6 [186].

#### 5.4.6. Effect of TQ on Janus Kinases-Signal Transducer and Activator of Transcription Factor (JAK-STAT)

The JAK/STAT pathway functionally modulates the TME. JAK/STAT signaling pathway and contains four JAK domains (JAK1,2,3, and tyrosine kinase 2 (TYK2)) and seven STAT proteins (STAT1-4, 5A, 5B, and 6) [187]. Dysfunction of the JAK/STAT pathway leads to tumorigenesis, metastasis, invasion, survival of tumor cells, proliferation, anti-apoptosis, angiogenesis, and immune invasion [188]. The JAK/STAT pathway in BC is altered by down-regulation of phophotyrosine-specific phosphatase in TNBC [189]. It is also impaired by activation of PI3K/mTOR [190] and down-regulation negative regulators of STAT, including suppressor of cytokine signaling 3 [191]. Increasing the amount of JAK/STAT results in activating ligand IL-6 [192]. Precise regulation of STAT activation is critical for eliciting the appropriate responses to extracellular signals [193], but persistent activation of STAT3 via phosphorylation by cytokines is mediated through JAK [194] and is required for the survival of tumor cells by promoting the upregulation of genes that encode anti-apoptotic proteins, angiogenic factors, and cell cycle regulators [193]_._

Targeting JAK/STAT activation alone can lead to unanticipated effects on the TME. JAK/STAT inhibition and reverses the negative impacts within the TME on immune cell function. Authors reported that JAK2-specific inhibitors had been shown in reducing tumor growth in TNBC cells, but JAK1/STAT3 inhibitor had little effect, sometimes counteracting the JAK2 specific inhibition on TNBC in an in vivo model [195]. 

There are no good studies about the impact of TQ on this target site; it will serve as a potential alternative site for the treatment/prevention of TNBC. Still, studies in liver cancer confirmed that TQ strongly suppressed the phosphorylation of STAT3 associated with decreased JAK activity, thus enhancing the apoptotic death rate of cancer cells [196]. TQ has also been shown in studies to reduce cell proliferation by blocking the activation of the JAK2/STAT3 signaling pathway in gastric cancer cells both in vitro (HGC27, BGC823, and SGC7901 cell lines) and in a xenograft tumor mouse model. TQ reduced the activation and phosphorylation of the STAT3 pathway and the production of STAT3-dependent reporter genes such as survivin, cyclin D, and VEGF, preventing cell proliferation and inducing apoptosis [197]. TQ inhibited STAT3 phosphorylation at Tyr705 due to JAK2 and Src activity suppression in gastric cancer cell line (HGC27) investigations. TQ inhibited STAT3 and prevented the expression of multiple STAT3-regulated genes, indicating that it is an efficient factor for blocking the STAT3 pathway by preventing tyrosine kinase phosphorylation [197]. Furthermore, TQ inhibited STAT3 activation and the production of STAT3-regulated gene products such as cyclin D1, Bcl-2, Bcl-xL, survivin, Mcl-1, and vascular endothelial growth factor in human multiple myeloma cell lines (U266 and RPMI 8226) [185]. Based on these findings, TQ will serve as a potential therapy option for TNBC by addressing this signaling system.

### 5.5. Effect of TQ on Endothelial Cells

#### 5.5.1. Effect of TQ on Vascular Endothelial Growth Factor-A (VEGF-A)

Vascular endothelial growth factor-A (VEGF-A), also referred to as vascular permeability factor, essential endothelial cells in angiogenesis and vasculo-genesis [198], is secreted by solid tumor for blood vessel formation to provide adequate blood to tumors for further stimulation of tumor proliferation and metastasis [199]. VEGF is highly involved in angiogenesis and increasing vascular permeability, and acts as a classical pro-tumorigenic NF-kB target gene. Overexpression of VEGF-A in BC is associated with reduced overall survival and disease-free survival mediated by endothelial growth factor receptors [200]. It is an essential molecular driver of highly aggressive cancers such as TNBC [201,202]. Upregulation of VEGF-A on endothelial cells increases cell survival, permeability, cell migration, and cell proliferation [203]. Parallel to the BC cell, it induces cell migration and cancer stem cell self-renewal; these two actions promote BC progression and metastasis [204]. VEGF is highly upregulated in the breast TME and activates oncogenic signaling pathways such as MAPK pathway and the phosphatidylinositol 3-kinase (PI3K)/Akt pathway, stimulating survival and proliferation, migration and angiogenesis [205]. 

Agents targeting VEGF help to prevent its interaction with vascular endothelial cells enabling increase apoptosis through decreasing cell proliferation, blood cell formation, and vascular permeability towards tumor cells [206]. TQ has confirmed anti-neoplastic activity via its ability to regulate several genetic pathways. In addition, TQ has established anti-cancerous effects by inhibiting the initiation, invasion, migration, and progression of cancer. TQ’s anti-cancer effects are chiefly mediated by regulating various cell signaling pathways such as VEGF, Bcl2/Bax ratio, p53, NF-kB, and other oncogenes [207]. Studies have shown that TQ down-regulated the expression of the VEGF-A gene and induced apoptosis in stomach cancer cells [208]. TQ considerably decreased VEGF-induced angiogenesis in vitro and in vivo, according to in vitro aortic ring assays and in vivo Matrigel plug assays. Furthermore, co-injection of TQ (6 mg/kg/day) for 15 days in a xenograft mice model with human prostate cancer cells (PC3 cells) resulted in a six-fold and 23-fold decrease in the size and weight of prostate tumors, respectively [209]. Another in-vitro study revealed that TQ down-regulates VEGF expression in hormone-positive BC cells [210]_._ Similarly, in vivo studies have shown that TQ, in combination with Resveratrol had a synergistic effect on induction of apoptosis anti-angiogenesis activity by decreasing expression of VEGF in mouse epithelial BC cells [211]. In line with this, TQ was also reported to inhibit the expression of NF-κβ-regulated genes VEGF-A in MDA-MB-231 TNBC cells [107]. These findings showed that TQ, which targets VEGF, could be a viable alternative in preventing and treating TNBC.

#### 5.5.2. Effect of TQ on Transforming Growth Factor-β (TGFβ)

Transforming growth factor-β (TGFβ) is a multifunctional polypeptide that acts as a tumor suppressor in normal epithelial cells and the early stage of tumor cells. In contrast, in an advanced stage of tumor cells, it acts as tumorigenesis [212]; TGF-β signaling is mediated by two transmembrane receptors, TGF-β type I and II, and the ligand binding leads to SMAD2/3 phosphorylation and activation [213]. TGF-β binding to its receptor leads to active-like kinase recruitment and propagation of downstream intracellular signaling pathways to the nucleus through Smad proteins that act as a transcription activator [214]. TGF-β promotes tumor cell invasion and migration by inducing an epithelial–mesenchymal transition. It is a profound phenotypic conversion by which epithelial cells lose their cohesiveness and polarity that acquire motile and invasive properties. It plays a major role in the morphogenetic process during organogenesis [215]^.^

In BC, TGF-kβ regulates several processes such as proliferation, differentiation, immunity, migration, and apoptosis. It has a dual function in the progression of tumors. It acts as tumorigenesis in the late stage that enhances invasion, migration, and survival of the tumor cells. A tumor suppressor in the early stage has an antiproliferative effect [216,217]. Authors have reported that TNBC is resistant to the growth-inhibitory activity of TGF-β [218]. In normal epithelial, endothelial, and neuronal cells, TGFβ has cytostatic effects and prevents hyper cell proliferation by controlling the G1/S phase transition events, mediated through cyclin-dependent kinase inhibitors [219,220]. The cell cycle arrest is also achieved by downregulation of the proliferative inducing transcription factor, c-Myc [221]; meanwhile, these antiproliferative effects can be opposing in an advanced stage. Evidence revealed that the TGF-β switches from tumor suppressor to pro-metastatic factor in BC progression [222].

The diversity effects of the TGF-β signaling pathway indicate a need for caution when targeting the design of effective TNBC therapeutics and/or prevention by identifying the factors that determine the balance between the opposing effects. It has been reported that TQ significantly reduced tumorigenic signaling such as TGF-β, with a consequence concentration-dependent inhibition of cancer cell growth, migration, and invasion [223]. According to the findings, TQ inhibited the metastatic phenotype and reversed EMT in prostate cancer cells in vitro by negatively modulating the TGF-/Smad2/3 signaling pathway [224]. These data imply that thymoquinone, which works by blocking TGF-β, could be a promising treatment agent for TNBC.

#### 5.5.3. Effect of TQ on Insulin-Like Growth Factor I (IGF-I)

Insulin-like growth factor I (IGF-I) is essential for fast cell growth, cell differentiation, survival, migration, and invasion in BC [225]. IGF-1 binds to insulin-like growth factor receptor-type1(IGF-1R) and stimulates various signaling pathways, promoting the growth and survival of cancer cells [226]. More importantly, the IGF-1/IGF-1R system mediates stimulatory effects in cancer cells via different routes such as the phosphatidylinositol-3kinasw(PI3K)/Akt1, mTOR, and MAPK [227,228]. Expression of IGF-1 activates MAPK and Akt, leading to increased cell proliferation, tumor cell survival, invasion, migration, tumor cell growth, epithelial to mesenchymal transition, and anti-apoptotic effects [229,230].

IGF-I is a well-known activator of the Janus-activated kinase/signal transducer and activator of transcription (STAT) pathway [231] and has been implicated in NF-κB–mediated transcriptional regulation of inflammatory cytokines and vascular endothelial cell adhesion receptors. IGF-I promotes TNBC metastasis not only through a direct paracrine effect on tumor cell survival and proliferation but also through indirect effects, likely involving the TME and pro-inflammatory response [232].

It has been reported that upregulation of IGF-1R and increased IGF-1 levels are associated with a high BC risk and poor prognosis in BC patients [233]; furthermore, in TNBC patients, it has been associated with shorter survival [234]. Stimulation of IGF-I protects BC from the toxic effects of chemo- and radiotherapy [235] as well as contributing to disease progression through its anti-apoptotic and mitogenic effects on the mammary epithelial cells [236]. Authors have reported that patients diagnosed with BC showed a high level of IGF-I detection in 100% of TNBC cases [237]. High IGF-1 gene expression signature mediating cancer proliferation and short survival in TNBC cells, and TQ targeting this site suggests a rationale for prevention and/or treatment. In vitro studies targeting the IGF-IR gene in normal and malignant human breast cancer tissue specimens have demonstrated promising anti-neoplastic effect [238]. TQ significantly augments cisplatin-induced anti-tumor effects on gastric cancer both in vitro and in vivo by blocking the PI3K/AKT signaling pathway, according to research done with SGC-7901 human gastric cancer cells and BALB/c mice implanted with gastric cancer cells [239]. These signals are IGF-1’s downstream pathways, suggesting that TQ could be employed as an alternative to IGF-1 targeting in the treatment and prevention of TNBC.

#### 5.5.4. Effect of TQ on Endoglin

Endoglin or CD105 is a homo-dimeric cell membrane glycoprotein receptor for TGF-β and bone morphogenetic protein-9(BMP-9), highly expressed in angiogenetic endothelial cells [240]. In addition to a co-receptor for TGF-β, endoglin is a proliferation-associated cell membrane antigen [241], indispensable for the development of angiogenesis [242], which promotes endothelial proliferation, and increases TGF-β/ALK1 (Activin receptor-like kinase 1) signaling transduction [243]. Since endoglin is expressed at a low level in quiescent endothelial cells, its expression is significantly upregulated in active vascular endothelial cells during tumor angiogenesis which directly involves cell proliferation, migration, capillary tube formation, and function as a pro-angiogenic role [244].

Overexpression of endoglin is strongly associated with metastasis, poor chemotherapy response, poor survival in BC patients, and the deletion of endoglin results in reversing carcinogenesis and chemoresistance [245,246]. Endothelial cells in the newly synthesized blood vessels in the TME significantly express the TGF-β co-receptor endoglin, which has been linked with poor prognosis and metastatic disease in many cancer cells, including TNBC [247].

Targeting endoglin and the VEGF pathway concurrently improves treatment in vitro and appears to reverse resistance in refractory cancer patients [248]. Studies targeting VEGF receptor kinase inhibitors as antiangiogenic therapies caused therapy resistance because of upregulation of another pro-angiogenic signaling pathway, such as the endoglin/transforming growth factor-β (TGF-β) pathway [249]. However, inhibiting VEGF and endoglin reduces tumor angiogenesis and decreases metastatic spread on BC cells [250]. It has been reported that agents that neutralize its antibodies, or a ligand trap targeting endoglin, inhibit BC metastatic spread and tumor angiogenesis in the mice model [250]. Preventing or blocking the tumor blood supply by disrupting the new blood cell formation has been considered as a general strategy to treat cancer [251]. TQ could be used to target TGF-B, VEGF, and other angiogenic factors in TNBC [158,209,252]. Targeting endoglin and its downstream pathways as an approach for treating TNBC with TQ would be another potential target site in this regard (Figure 6).

#### 5.5.5. Effect of TQ on Indoleamine 2,3-dioxygenase (IDO)

Indoleamine 2,3-dioxygenase (IDO) is an immunocompromised enzyme that degrades tryptophan, essential for activation of T-cell, leading to inhibition of T-cell proliferation [253]. Following the catabolism of tryptophan, it generates different physiological active immunosuppressive metabolites such as kynurenine [254]. Kynurenine causes tumor-infiltrating lymphocytes to become anergic and die [255]. Blocking activation of T-cell reduces its prominent role in modulating host response to tumor cells and plays an immunosuppressive role following inflammatory stimuli and preventing immune activation [256]. Upregulation of IDO results in immunotolerance and immunosuppression and can obstruct the antitumor responses and enhance metastasis in BC [257]. IDO is expressed by macrophages [253], trophoblasts, and dendritic cells [258]. IDO expression in the TME has been correlated with the induction of multiple tolerogenic immune phenotypes, including the inhibition of effector T cell activation, enhanced infiltration of myeloid-derived suppressor cells, B cell dysfunction, and promotion of tumor angiogenesis [259].

Evidence suggests that various types of human tumor cell, including TNBC, express IDO, and inflammatory mediators, particularly interferon-*γ* (IFN*γ*), have the specific ability to enhance the expression of IDO [260]. Studies reported a high level of IDO expression in primary tumors and serum of BC patients [261]. Even so, further investigation is suggested, as it is not expressed in normal breast tissues [262]. TNBC cells express IDO in the presence of inflammation and T-cell infiltration [263]. Evidence from Pakistan showed that from a total of 100 BC patients, high IDO was observed in TNBC patients compared to non-TNBC patients, at 65.3% and 33.3%, respectively, and this overexpression of IDO was associated with decreased overall patient survival [264]. IDO expression has been shown to correlate with poor clinical prognosis in various cancers [265,266,267]. IDO in tumor cells can potentially suppress T-cells during tumor growth; it reduces not only the amount of T-cell infiltration but also compromises the cytotoxic function of effector CD8^+^ T-cells [268].

Since the immune system in BC patients is dysfunctional [269], targeting the IDO site is an alternative option for repopulating the host with immune cells, relieving IDO-mediated immune suppression in the TME and tumor cells themselves [270]. These inhibitory effects might converge to induce cytotoxic T-cell exhaustion and dampen antitumor immunity. It has been reported that IDO inhibitor combine with taxane augmented tumor-infiltrating lymphocytes to kill tumor cells and improve clinical outcomes in BC patients [271], since pro-inflammatory molecules such as IL-1, IL-6, and IFN induce expression of IDO through STAT independent pathway involving P38, MAPK, and NF-kβ [272] and these cytokines and signaling pathways are inhibited by TQ [185,273,274], so that IDO will be another potential target site for TQ.

## 6. Clinical Trials Have Shown the Importance of TQ in the Treatment of a Variety of Diseases

We highlighted the importance of TQ in the treatment of various diseases in various countries based on completed clinical trials, despite the fact that there are no published clinical trials of TQ on TNBC in the United States [275] (Table 1).

## 7. Limitations of Thymoquinone (TQ) as a Natural Product

One of the most pressing concerns is TQ’s bioavailability. According to pharmacokinetic studies, TQ is rapidly eliminated and slowly absorbed, resulting in a decreased bioavailability of roughly 58% and a lag time of 23 min [282]. Several studies have been conducted to boost bioavailability by changing the chemical structure of the medicine and generating nanoparticles, and have proven to be successful. A thymoquinone-loaded nanostructured lipid carrier was devised in cancer cell lines to improve bioavailability (elimination half-life of 5 h) and induction of apoptosis and cell cycle arrest [283,284]. The investigators developed thymoquinone-encapsulated nanoparticles using biodegradable, hydrophilic polymers such as polyvinylpyrrolidone and polyethylene glycol to overcome thymoquinone’s poor solubility, thermal and light sensitivity, and minimal systemic bioavailability, which can significantly improve the cancer treatment’s efficiency. This nanoparticle can kill and inhibit the migration of breast cancer cells [285].

Another concern with clinical TQ is the issue of safety. Drugs are used to treat several degenerative diseases; however, many drugs are limited by side effects and toxicity. In many cases, bioactive compounds of eatable medicinal plants have a promising and prominent role in human health due to low toxicity [286], as with supplementation of TQ [287]. Studies indicate that TQ is not toxic in rats within the range of 10–100 mg/kg body, both in sub-chronic and subacute doses [288]. The LD50 of TQ was estimated to be 104.7 mg/kg after intraperitoneal injection and 870.9 mg/kg after oral intake in mice, and 57.5 mg/kg and 794.3 mg/kg in rats, respectively, following intraperitoneal and oral administration [289]. In another study, investigators revealed no side effect of thymoquinone on prolonged consumption of TQ (10–100 mg/kg up to 20 weeks) [290]. The acceptable dose of TQ for humans is 0.6 mg/kg/day [291]. On the other hand, TQ has anticancer action at relatively low concentrations, around 10 mg/kg [292] As a result, the issue of safety may not be a major concern.

## 8. Conclusions

TNBC is not simply a group of cancer cells, but rather a heterogeneous collection of various compounds that support the growth and progression of tumor cells. An emerging TME is quite a complicated and continuously evolving topic in the research area. We would like to focus on selected components in the microenvironment that directly or indirectly contributed to TNBC cell growth, proliferation, migration, invasion, and development of drug resistance. The composition of the TME is bulk and conventional therapy, even in combination, is not effective in the treatment of TNBC cases. The characteristics of the TME can be manipulated to design more effective therapies. TQ has been reported to down-regulate various tumorigenic induced associated signals and proteins. This review described essential features of the TME components and discussed strategies whereby TQ may be effective. Hence all these data suggest that TQ alone or in combination with an existing drug, focusing on the cancer-producing mechanisms of each TME component signaling process stage, can be a potential alternative for the success of therapeutic intervention against TNBCs. Since various components of the TME serve as a favorable shelter for TNBC cells, TQ will be helpful in prevention of TNBC through the modulation of the tumor microenvironment.

## Figures and Tables

**Figure 1 nutrients-14-00079-f001:**
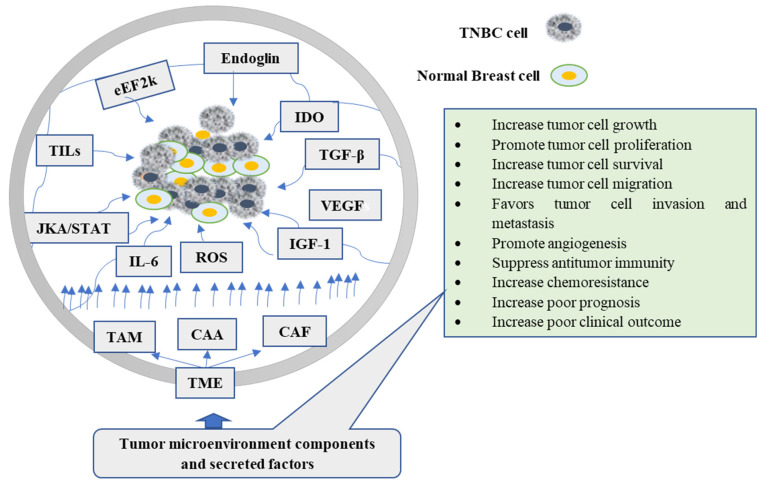
The role of tumor microenvironment components and their secreted factors in the carcinogenesis of TNBC cells. TAM, CAA, and CAF are the major cellular components of TME, while also shown in the figure are secreted factors and signals. CAA: Cancer-associated adipocytes; CAF: Cancer-associated fibroblast; eEF2k: eukaryotic elongation facto 2k; TAMs: tumor-associated macrophages; TGF-β: transforming growth factor-β; IGF: insulin-like growth factor; ROS: reactive oxygen species; IL-6: Interleukin-6; TNBC: triple-negative breast cancer; VEGF: Vascular endothelial growth factor; IDO: Indoleamine 2,3-dioxygenase.

**Figure 2 nutrients-14-00079-f002:**
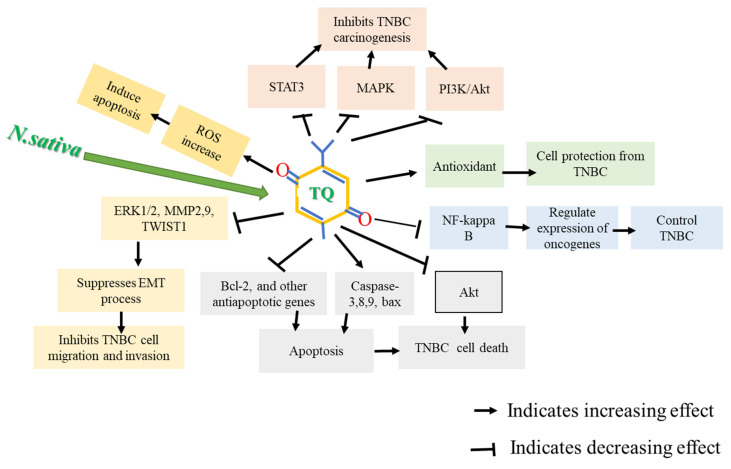
Possible anti-cancer mechanism of thymoquinone (TQ). TQ caused apoptosis in TNBC cells by inducing apoptotic genes, downregulating anti-apoptotic genes, and generating reactive oxygen species (ROS). TQ also suppresses Akt activation, deactivates Nf-kB, and increases antioxidant enzymes, inhibiting TNBC carcinogenesis.

**Figure 3 nutrients-14-00079-f003:**
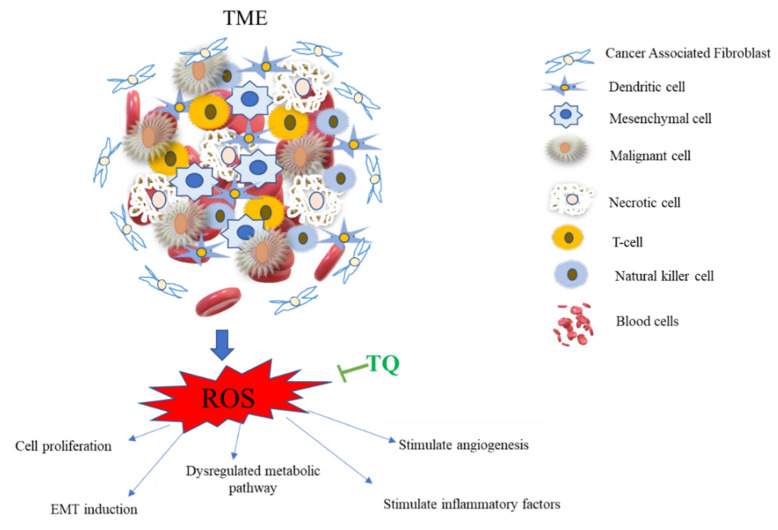
TQ inhibition of reactive oxygen species (ROS) production in the tumor microenvironment (TME). The TME is a complicated web of inflammatory and non-inflammatory cells. ROS boosts cancer cell proliferation, EMT stimulation, metabolic system dysregulation, and angiogenesis stimulation through altering vascular endothelial growth factors like VEGF and promoting inflammatory chemicals. The metabolic processes of glycolysis and oxidative phosphorylation are strictly controlled by ROS, with an excess of ROS resulting in a dysregulated metabolic pathway [93]. TQ counteracts all of these effects by inhibiting the generation of reactive oxygen species (ROS) via activating cytoprotective enzymes and nuclear factor-erythroid 2 related factor 2 (Nrf2).

**Figure 4 nutrients-14-00079-f004:**
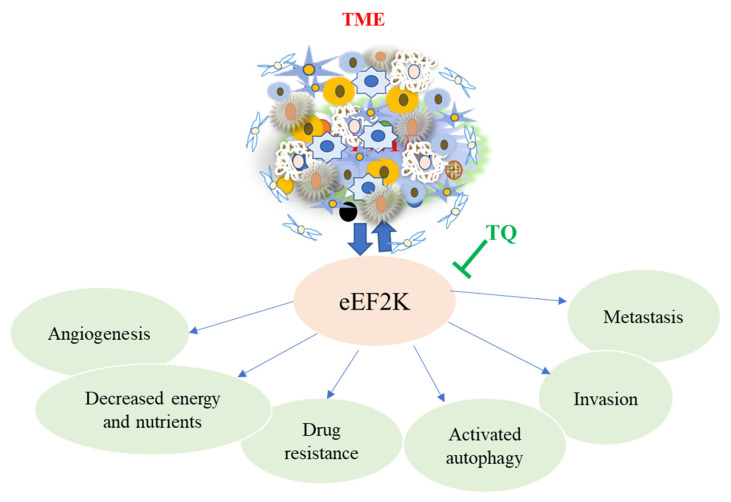
The major role of eEF2K in promoting angiogenesis, metastasis, invasion, and drug resistance in cancer cells, including TNBC. These effects promote cancer progression, resulting in tumor growth, poor prognosis, and short survival in TNBC patients. TQ reverses this action by regulating and inhibiting the upregulation of eEF2K in the TME.

**Figure 5 nutrients-14-00079-f005:**
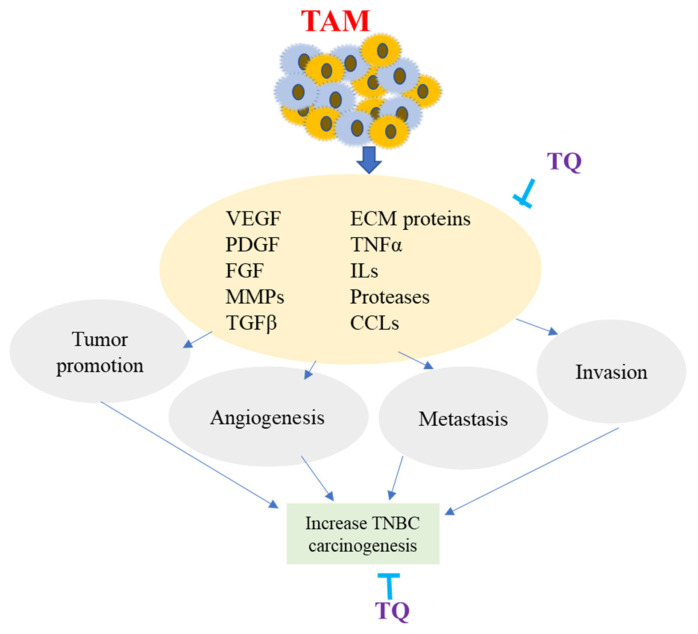
Overview of Tumor-Associated Macrophage (TAM). TAM is produced from monocytes recruited at the tumor site by molecules produced by TME. Major factors involved in TAM are the chemokines (CCLs), vascular endothelial growth factor (VEGF), interleukins (ILs), transforming growth factor (TGFa), matrix metalloproteinases (MMPs), extracellular matrix proteins (ECM), and platelet-derived growth factors (PDGF). TAM produces several molecules that sustain malignant cell survival, modify neoplastic ECM proteins, enhance invasion and metastasis, promote the development of a newly formed vessel, and assist tumor cells in their progression. TQ potentially inhibits TAM-related expression of various factors and proteins, resulting in decreasing angiogenesis, invasion, and metastasis of TNBC.

**Figure 6 nutrients-14-00079-f006:**
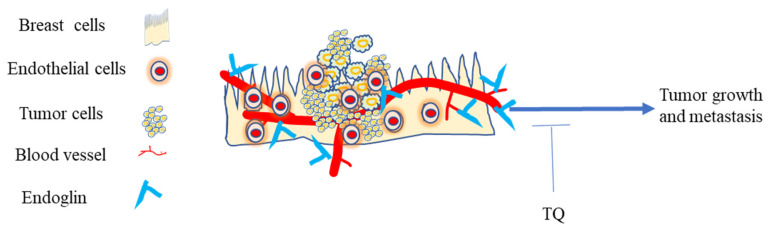
Endoglin in the TME of TNBC. The expression of endoglin on endothelial cells is critical for promoting primary tumor growth and metastasis. Blocking endoglin in the TNBC TME with TQ exerts anti-cancer activity.

**Table 1 nutrients-14-00079-t001:** Summary of TQ application in clinical trials for various diseases.

Disease	Study Type	Major Clinical Findings	Ref.
Intractable pediatric seizures	Double-blinded crossover clinical trial	The frequency of seizures decreased significantly with extract of *Nigella sativa*	[276]
Advanced Refractory Malignant Disease	Open-label, non-randomized cohort study	TQ had no adverse effects and no anti-cancer effects	[277]
Polycystic Ovary Syndrome (PCOS)	Randomized clinical trial	Supplementing *black cumin oil* with metformin improves PCOS-related symptoms	[278]
Type II Diabetes millets	Randomized double-blind, placebo-controlled trial	Fasting blood sugar, glycated hemoglobin, triglyceride, and low-density lipoprotein–cholesterol levels all changed considerably in the intervention group (*Nigella sativa*) compared to the placebo group.	[279]
Ulcerative colitis	Prospective, randomized, double-blind, placebo-controlled trial	No significant difference between the two groups (Placebo vs. *Nigella sativa*)	[280]
Asthma disease	Double-blind, placebo-controlled trial	All asthma symptoms, frequency of asthma symptoms/week, chest wheezing, and pulmonary function tests values in the study group (*Nigella sativa*) had significantly improved	[281]

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
