# Peer review of "Therapeutic Potential of Thymoquinone in Triple-Negative Breast Cancer Prevention and Progression through the Modulation of the Tumor Microenvironment"

_nutrients, 2021, doi:10.3390/nu14010079_

Round 1

Reviewer 1 Report

In this review, the authors focused on the description of the tumour microenvironment and the search for targets for thymoquinone (TQ). They first described the tumour microenvironment, giving some data on natural products and TQ (lines 58-145). Then the rest of the review focuses on the study of the different targets of the microenvironment: Cholesterol synthesis, Reactive oxygen species, eEF-2K, Inflammatory and immune cells (Tumor-Associated Macrophage, Cancer-associated adipocytes, Cancer-associated fibroblasts, Tumor-infiltrating lymphocytes, IL-6, JAK-STAT), Endothelial cells.

Major comments:

The title is “Therapeutic Potential of Thymoquinone in TNBC Prevention and Progression Through the Modulation of the Tumor Micro-environment”. Contrary to what is indicated in the title, this review does not really focus on TNBC. In view of the Pubmed literature, the authors should have focused more on the available data specific to TNBC.

Concerning TQ, the authors sometimes give some data on the mechanisms of action of TQ. but most of the time, the authors simply say that TQ "could act" on the targets described. again, this review would deserve a thorough analysis of the data in the literature.

Minor comments:

Title : write “TNBC” not in abbreviated form

Lines 35-37: please modify the reference n°1 and put a reference which includes epidemiological data. Rewrite the first sentence

Line 54: “Given that TNBC is one of the most common,”, TNBC is not the most common…

Line 65: supress “in”

Line 68: add cancer “cells”

Figure 1: the figure must be revised to show the presence of CAF, CAA in the TME. The figure must show the difference between secretions and cells. Information is given in bulk

Line 153 : “reactive oxygen specious »

Line 213 : « So, they are recognized as the hallmarks of cancer include the tumor-promoting microenvironment” : must be rewriten

Figure 2: it will be necessary to delete the arrow which inhibits the TME because it is too general. just leave the inhibition of the ROS

Line 250: “both in vivo and in vitro models(82).” : references should be added to cover in vitro and in vivo studies

Line 305 “The role of inflammatory/immune cells of the TME by increase cancer cell”: supress “by”

Line 305-308: presence of repetition. the sentence should be rewritten

Lines 352-355: the authors say "recent studies" when they only cite one from 2007...

Figure 5: too simplistic

Lines 392-395: sentence to be rewritten because IL6, TNF are adipokines.

Paragraph 5.4.2: there is no data on the effect of TQ on CAAs. This paragraph give non information on TQ

In paragraph 5 (lines 151-690) which corresponds to almost the entire manuscript, most of the text corresponds to the description of the cells of interest (TAM, CAA, etc...). As regards to TQ, too little description of the effects is given (4-5 lines). For example, in paragraph 5.4.1, only 4 lines correspond to the effects of TQ. the rest is devoted to tam

There is no real data on TQ. For example, paragraph 5.4.4 (lines 432-449): authors describes TILs and concluded “These data suggested that high TILs are helpful for better survival and predict a positive response to chemotherapy, indicating that TQ will be important in this point of view for TNBC patients.”

this is general throughout the manuscript.

Why make a paragraph on IL6 when it is produced by TAM and CAA?

There are a lot of generalities and it is not too focused on TNBC

For example, paragraph 5.4.1, it is data that is as valid for TNBC as for other types of breast cancer

Author says that TQ could target this or that pathway without real scientific data

In general, it should be specified in which models the effects are described. For example, in vitro: on which cancer lines, in vivo animals or clinics so that the role of TQ can be clearly seen

Author Response

Dear editor:

We are pleased to resubmit the revised version of the Manuscript ID:  nutrients- 1489833. Title: " Therapeutic Potential of Thymoquinone in Triple-Negative Breast Cancer Prevention and Progression Through the Modulation of the Tumor Microenvironment." We appreciate the reviewers' constructive criticisms, questions, and comments, and we have addressed each of their concerns as outlined below.

Reviewer 1

Major comments:

  1. The title is "Therapeutic Potential of Thymoquinone in TNBC Prevention and Progression Through the Modulation of the Tumor Microenvironment." Contrary to what is indicated in the title, this review does not really focus on TNBC. In view of the Pubmed literature, the authors should have focused more on the available data specific to TNBC.

Response: Corrected as suggested. We revised the whole manuscript based on the suggestions

  1. Concerning TQ, the authors sometimes give some data on the mechanisms of action of TQ. but most of the time, the authors say that TQ "could act" on the targets described. again, this review would deserve a thorough analysis of the data in the literature. More data are included as suggested

Response: Corrected as suggested. We revised the whole manuscript based on the suggestions

Minor comments:

Title: write "TNBC" not in abbreviated form-

Response: are revised based on the comment as follows and was included in the main text (lines 2 and 3) "Therapeutic Potential of Thymoquinone in Triple-Negative Breast Cancer Prevention and Progression Through the Modulation of the Tumor Microenvironment."

3 Lines 35-37: please modify the reference n°1 and put a reference which includes epidemiological data. Rewrite the first sentence-

Response: modified as suggested by including local and global epidemiological data and included in the main text. (Line 37 and 38) "In 2020, there were approximately 2.3 million new breast cancer cases and 685,000 breast cancer deaths globally, with 1,898,160 new cases and 608,570 deaths projected in the United States by the end of 2021."

4 Line 54: "Given that TNBC is one of the most common," TNBC is not the most common…

Response: corrected as suggested and included in the main text. (Line 54,55) "Given that TNBC is one of the most common life-threatening and complex kinds of BC, substantial research has been devoted to the discovery of novel biomarkers and biologically targeted treatment sites in order to enhance patient prognosis and clinical outcomes, in comparison to other subtypes."

5     Line 65: suppress "in"

 Response: modified as suggested and included in the main text. (line 64). "The TME generates a place for dwelling and interacting cancer cells with their neigh-boring immune and endothelial cells as well as fibroblasts and can provide potential targets for TNBC."

6     Line 68: add cancer "cells"-

Response: corrected as suggested and included in the main text. (line 67) The reciprocal communication between stromal cells and cancer cells produces changes in the cellular elements of TME, which predisposes tumor cells to metastasis

7     Figure 1: the figure must be revised to show the presence of CAF, CAA in the TME. The figure must show the difference between secretions and cells. Information is given in bulk.

Response: corrected as suggested and included in the caption as follows and was included in the main text (Lines 84, 85). The role of tumor microenvironment components and its secreted factors on the carcinogenesis of TNBC cells. TAM, CAA, and CAF are the major cellular components of TME, while the rest shown in the figure are secreted factors and signals. CAA: Cancer-associated adipocytes; CAF: Cancer-associated fibroblast; eEF2k: eukaryotic elongation facto 2k; TAMs: tumor-associated macrophages; TGF- β: transforming growth factor-β; IGF: insulin-like growth factor; ROS: reactive oxygen species; IL-6: Interleukin-6; TNBC: triple-negative breast cancer; VEGF: Vascular endothelial growth factor; IDO: Indoleamine 2,3-dioxygenase.

8     Line 153: "reactive oxygen species.»

Response: corrected as suggested as reactive oxygen species as follows and was included in the main text (line 141). The cellular components and secretary molecules of TME targeted by TQ as discussed here are tumor-associated macrophage(TAM), cancer-associated fibroblast(CAF), cancer-associated adipocytes(CAA), cholesterol synthesis, reactive oxygen species (ROS), eukaryotic elongation factor-2 kinase (eEF-2K), tumor-infiltrating lymphocytes (TILs), indoleamine 2,3-dioxygenase (IDO), vascular endothelial growth factor (VEGF), transforming growth factor-β (TGF-β), insulin-like growth factors (IGF-I), endoglin, IL-6, and JAK/STAT signaling.

9     Line 213: « So, they are recognized as the hallmarks of cancer include the tumor-promoting microenvironment": must be rewritten-

Response: corrected as follows and was included in the main text (line 218) "As a result, ROS have been identified as a distinguishing feature of cancer cells secreted in the tumor-promoting microenvironment."

10   Figure 2: it will be necessary to delete the arrow which inhibits the TME because it is too general. just leave the inhibition of the ROS-

Response: This is figure 3 and is corrected as suggested as follows and was included in the main text - (line 271)

11   Line 250: "both in vivo and in vitro models (82).": references should be added to cover in vitro and in vivo studies

Response: corrected as suggested- added two more references as follows and was included in the main text - line 257 "Meanwhile, TQ has dual functions towards the regulation of ROS. Evidence shows that TQ inhibits proliferation and promotes apoptotic effects through upregulation of ROS in BC cells both in vivo and invitro models."

12   Line 305 "The role of inflammatory/immune cells of the TME by increase cancer cell": suppress "by"

Response: rewritten as suggested as follows and was included in the main text -line 320 "Inflammatory/immune cells in the TME play a role in cancer cell survival, development, proliferation, and differentiation by boosting growth and survival factors and encouraging angiogenesis, inducing mutation by damaging DNA, and interfering with treatment."

13   Line 305-308: the presence of repetition. the sentence should be rewritten-

Response: rewritten as suggested as follows and was included in the main text -line 320-323 "Inflammatory/immune cells in the TME play a role in cancer cell survival, development, proliferation, and differentiation by boosting growth and survival factors and encouraging angiogenesis, inducing mutation by damaging DNA and interfering with treatment."

14   Lines 352-355: the authors say "recent studies" when they only cite one from 2007...

Response: corrected as suggested and included in the main text. Line 382-385 Furthermore, in an orthotopic breast tumor model, anti-VEGF-A neutralizing antibody therapy reduced the creation of new blood and lymphatic vessels, as well as the incidence of lymphatic and pulmonary metastasis. VEGF promotes macrophage recruitment into tumors, and studies demonstrated that specific inhibition of VEGFR2 decreases tumor macrophage infiltration significantly.

15   Lines 392-395: sentence to be rewritten because IL6, TNF are adipokines.

Response: rewritten as suggested and included in the main text. line 440-441 "It has been reported that adipokines produced by adipocytes, which secrete ILs, TNF-α, and hepatic growth factor, support the growth of BC by transferring fatty acids to enhance mitochondrial -oxidation, extracellular matrix remodeling, and endotrophin production from collagen, support the growth of BC by transferring fatty acids to enhance mitochondrial -oxidation, extracellular matrix remodeling, and endotrophin production from collagen"

16   Paragraph 5.4.2: there is no data on the effect of TQ on CAAs. This paragraph gives non-information on TQ-

Response: Data included as suggested and was included in the main text paragraph 5.4.2 line 453-463. TQ has been shown to have anti-inflammatory and immunomodulatory properties in both cellular and humoral immunity, as well as antiangiogenic effects. According to a comprehensive review of studies, TQ has also been proven to protect against inflammatory factors by lowering proinflammatory cytokines like IL-1, IL-6, and TNF-α, which are also produced by CAAs. Because thymoquinone reduces the synthesis of these damaging proinflammatory cytokines, it also reduces the creation of free radicals in the TME. Furthermore, in vivo animal investigations demonstrated that TQ reduced adipocyte hypertrophy-induced inflammation, implying that TQ could be an alternative option to prevent TNBC associated with cancer-induced adipocytes in the microenvironment.

17   In paragraph 5 (lines 151-690) which corresponds to almost the entire manuscript, most of the text corresponds to the description of the cells of interest (TAM, CAA, etc...). As regards TQ, too little description of the effects is given (4-5 lines). For example, in paragraph 5.4.1, only 4 lines correspond to the effects of TQ. the rest is devoted to tam

Response: more data regarding TQ included as suggested as follows and was included in the main text. Paragraph 5.4.1 line 387-400 "TQ has been shown to have considerable anti-inflammatory effects on macrophages after oral administration by suppressing inducible nitric oxide synthase enzymes in response to inflammatory stimuli in mice. According to research, TQ lowered tumor NF-β activity, M2 macrophages, and soluble VEGF levels, which reduced proliferation and enhanced apoptosis in the ID8-NGL mouse model of ovarian cancer. TQ's antiangiogenic activities were tested in a zebrafish angiogenesis model in vivo. The antiangiogenic action of TQ was examined in zebrafish embryos by testing its inhibitory influence on blood vessel formation followed by identification of its molecular target. TQ decreased the formation of intersegmental vessels in zebrafish embryos in a dose-dependent manner, according to the findings. TQ was also discovered to suppress VEGF-A mRNA expression. TQ has also been demonstrated to reduce cell proliferation, migration, invasion, and metastasis in TNBC by downregulating TAM-related chemokine receptor expression. TQ may have an anticancer effect via modifying proteins that are either overexpressed or underexpressed by TAMs, indicating that it could be used to prevent and treat TNBC, according to the findings."

18   There is no real data on TQ. For example, in paragraph 5.4.4 (lines 432-449): the authors describe TILs and conclude, "These data suggested that high TILs are helpful for better survival and predict a positive response to chemotherapy, indicating that TQ will be important in this point of view for TNBC patients."

Response: more data regarding TQ included as suggested as follows and was included in the main text. Paragraph 5.4.4 line 516-530. "There is no information on how TQ affects TIL in TNBC. However, some research have shown that TQ improves the immune system by improving the immunological response to malignancies. T lymphocyte cell depletion was detected in rats after exposure to gamma radiation, which was recovered by the injection of TQ in mice by raising the percentage of CD4+, CD8+ cells through modifying Bcl-2, Bax, PD-1, and caspase-3 expression. When compared to the placebo or baseline group, N. Sativa oil alleviated autoimmune disease by boosting the expression of CD4+ cells in a clinical trial. TQ may be engaged in anti-inflammatory effects in TNBC patients by increasing the activity of FoxP3+ Treg cells. TQ dramatically reduced diabetes complications in Streptozotocin-induced gestational diabetic mice by boosting T-cell proliferation and restoring thymus homing CD4+, CD8+, and circulating T-cells. Foxp3+, CD3+, CD4+, and CD8+ T lymphocytes are the broadly studied major components of TILs. Several pre-clinical and clinical trial studies supported those high levels of TILs present at diagnosis were strongly associated with reduced distant recurrence rates and favorably altered the immune microenvironment. This finding backs up TQ's immunomodulatory properties in TNBC patients, suggesting that TQ will be important to increase TILs through stimulating their core components such as Foxp3+, CD3+, CD4+, and CD8+."

19   This is general throughout the manuscript.

Why make a paragraph on IL6 when it is produced by TAM and CAA?-

Response: corrected as suggested as follows and was included in the main text. There is a subsection on IL6 and minimized its content on TAM and CAA.  paragraph 5.4.5

20   There are a lot of generalities, and it is not too focused on TNBC-

Response: corrected as suggested. Included more specific data on TNBC

21   For example, paragraph 5.4.1 is data that is as valid for TNBC as for other types of breast cancer. Author says that TQ could target this or that pathway without real scientific data

In general, it should be specified in which models the effects are described. For example, in vitro: on which cancer lines, in vivo animals, or clinics so that the role of TQ can be clearly seen

Response: are revised based on the comment as follows and was included in the main text

Reviewer 2 Report

Revision manuscript Nutrient-

The authors of the review entitled “Therapeutic Potential of Thymoquinone in TNBC Prevention 2 and Progression Through the Modulation of the Tumor Micro- 3 environment” deal with an interesting and well-debated topic regarding this kind of solid tumor. The topics are well treated and well correlated with each other and the figures are well done and understandable.

Just a brief paragraph, to improve the review: the authors should discuss the limitations in the pharmacological use of this natural molecule such as its bioavailability that in the specific case of the tumor microenvironment is a fundamental issue. Finally, the authors should add a table resuming in vivo studies or human clinical trials (if any) in progress or completed.

Author Response

Dear editor:

We are pleased to resubmit the revised version of the Manuscript ID:  nutrients- 1489833. Title: " Therapeutic Potential of Thymoquinone in Triple-Negative Breast Cancer Prevention and Progression Through the Modulation of the Tumor Microenvironment." We appreciate the reviewers' constructive criticisms, questions, and comments, and we have addressed each of their concerns as outlined below.

Reviewer 2

The authors of the review entitled "Therapeutic Potential of Thymoquinone in TNBC Prevention 2 and Progression Through the Modulation of the Tumor Microenvironment" deal with an interesting and well-debated topic regarding this kind of solid tumor. The topics are well treated and well correlated with each other, and the figures are well done and understandable.

Just a brief paragraph, to improve the review: the authors should discuss the limitations in the pharmacological use of this natural molecule, such as its bioavailability that in the specific case of the tumor microenvironment is a fundamental issue.

Response: are revised based on the comment as follows and was included in the main text

Line 783-809. Limitations of Thymoquinone (TQ) as a natural product. One of the most pressing concerns is TQ's bioavailability. According to pharmacokinetic studies, thymoquinone is rapidly eliminated and slowly absorbed, resulting in a decreased bioavailability of roughly 58 percent and a lag time of 23 minutes. Several studies have been conducted to boost bioavailability by changing the chemical structure of the medicine and generating nanoparticles, and they have proven successful in doing so. A thymoquinone-loaded nanostructured lipid carrier was devised in cancer cell lines to improve bioavailability (elimination half-life of 5 hours) and induction of apoptosis and cell cycle arrest. The authors developed thymoquinone-encapsulated nanoparticles using biodegradable, hydrophilic polymers like poly-vinylpyrrolidone and polyethylene glycol to overcome thymoquinone's poor solubility, thermal and light sensitivity, and minimal systemic bioavailability, which can greatly improve the cancer treatment's efficiency. This nanoparticle can kill and inhibit the migration of breast cancer cells. Another concern with clinical thymoquinone use is the issue of safety. Drugs are used to treat several degenerative diseases; however, many drugs are limited by side effects and toxicity. In many cases, bioactive compounds of eatable medicinal plants have a promising and prominent role in human health due to low toxicity, as in the case of supplementation of TQ. Studies indicate that TQ is not toxic in rats within the range of 10–100 mg/kg body, both in sub-chronic and subacute doses. The LD50 of TQ was estimated to be 104.7 mg/kg after intraperitoneal injection and 870.9 mg/kg after oral intake in mice, and 57.5 mg/kg and 794.3 mg/kg in rats, respectively, following intraperitoneal and oral administration.  In another study, investigators revealed no side effect of thymoquinone on prolonged consumption of TQ (10–100 mg/kg up to 20 weeks). The acceptable dose of TQ for humans is 0.6 mg/kg/day. On the other hand, TQ has anticancer action at relatively low concentrations, around 10 mg/kg. As a result, the issue of safety may not be a major concern.  Included on section related to the compound limitations.

Finally, the authors should add a table resuming in vivo studies or human clinical trials (if any) in progress or completed.

Response: Not incorporated in the review since there are no clinical trials of TQ in Breast cancer studies registered in the USA, which can be accessed at https://clinicaltrials.gov/ct2/results?term=Thymoquinone